# The Measurement Model of Family Strengths for Generation Alpha in the Thai Context

**DOI:** 10.3390/bs14100921

**Published:** 2024-10-10

**Authors:** Kanchana Pattrawiwat, Pitchaya Nilrungratana, Anusara Deewai, Sudarat Tuntivivat

**Affiliations:** Behavioral Science Research Institute, Srinakharinwirot University, Bangkok 10110, Thailand; kanchana.pattrawiwat@gmail.com (K.P.); pitchayan@g.swu.ac.th (P.N.); anusarade@g.swu.ac.th (A.D.)

**Keywords:** family strengths, generation alpha, exploratory factor analysis, confirmatory factor analysis

## Abstract

The concept of the Thai family has shifted from the traditional family to families of various forms, such as single-parent families, same-sex families, skipped-generation families, and one-person households, which affects not only mental health, but future relationships of the children. This research aimed to analyze the factors that contribute to family strengths in the context of Generation Alpha and develop a measurement model tailored to a Thai context. The sample consisted of parents and caregivers of Generation Alpha, divided into two sections, with 521 participants for exploratory factor analysis and 508 participants for confirmatory factor analysis. Using stratified random sampling, data were collected via questionnaires distributed across four regions of Thailand. The discrimination index ranged from 0.280–0.896, with a Cronbach’s alpha of 0.952. The results revealed a measurement model consisting of five key factors contributing to family strengths. The factor with the highest score was positive relationships, with a factor loading of 0.92. Additional factors included technology and media literacy, parental roles, good attributes, and mental immunity, with a factor loading of 0.80, 0.74, 0.55, and 0.44, respectively. The measurement model of family strengths for Generation Alpha in the Thai context was found to be consistent with the empirical data. These findings suggest that understanding these family strengths can significantly enhance the well-being of both Generation Alpha and their family members.

## 1. Background

Thailand has a deep-rooted tradition of valuing family and prioritizing the wellbeing of its younger generations. However, Thai children are experiencing significant transformations in family dynamics, which may affect their development [1]. Since 2011, Thai children have declined in several measurements, including Intellectual Intelligence (IQ) [2]. The average IQ score has dropped from 94 to 93, indicating that Thai children still fall short of the global average of 100. Both IQ and Emotional Quotient (EQ) scores are closely linked with the quality of familial relationships. If parents and caregivers spend more time engaging with children, they will be more likely to develop the cognitive and affective abilities that raise IQ and EQ scores. Despite this, the latest OECD Programme for International Student Assessment (PISA) has revealed that Thai students rank below average in mathematics, science, and reading (OECD, 2023). Moreover, Thai children are becoming increasingly susceptible to online threats [3].

The intergenerational relationships with families are crucial for the well-being of young generations [4]. Generation Alpha, encompassing those born between 2010 and 2024, is entering the world with unique experiences and challenges compared to previous generations [5]. Growing up with technology, Generation Alpha navigate a world of constant connectivity and information overload [6]. While this offers incredible opportunities, it also presents complex challenges concerning family dynamics [7].

It is undeniable that familial factors such as parental behavior, education, and income significantly influence the development of Thai children, especially Generation Alpha [8,9]. Additionally, there has been a notable shift from more traditional family structures to a more diverse array of family forms, such as single-parent families, same-sex families, skipped-generation families, and one-person households [10]. The rising prevalence of single-parent families and skipped-generation families has implications not only for mental health, but also for the future relationships of children [11]. While Thai family styles are becoming more diverse, issues such as domestic violence remain persistent, leading to long-term negative impacts on the education, health, and life satisfaction of individuals especially Generation Alpha, which exacerbates serious social problems [12].

Many studies on children and family have predominantly focused on deficit-based models of social practice, emphasizing the adversities faced by children and families [13]. This strength-based approach offers a positive and empowering lens for children and family development [14]. It also shifts away from the traditional position of problem orientation and risk-focused methodologies, and aims to promote well-being through positive engagement among family members [15]. By highlighting family strengths rather than individual pathologies, this approach fosters a more constructive view of familial relationships. When applied within the rich cultural context of Thai families, it can cultivate thriving environments, nurture strong relationships, and unlock the full potential of all members.

The concept of family strengths is multifaceted. encompassing qualities that enhance family resilience, support, and the ability to thrive in the face of challenges [16]. Family strength serves as a protective factor that assists young members in navigating through various crises caused by both external and internal pressures [17]. Importantly, family strength does not imply the existence of a perfect family dynamic or the attainment of some idealized image; rather, it centers on fostering a foundation of trust, love, and mutual support that enables every member to flourish [18].

According to the literature review, there is a lack of research on the factors contributing to family strengths within the Thai context, particularly concerning Generation Alpha. Although there have been a few studies about family strength, namely addressing the factors of family strength [19], indicators of family strength [20], indicators of strong families [21], second-order components of Thai family strength indicators [22] and the construction of family strength factors [23], comprehensive exploration remains limited. Therefore, this research aims to investigate the factors influencing family strengths and to develop a measurement model specifically tailored for Generation Alpha in the Thai context. The findings of this research can be used as a guideline for family strength for Generation Alpha in the Thai context, who are growing up in a world of advanced artificial intelligence, digital connectivity, and dynamic environment which significantly influences their socialization and family dynamics differently from previous generations which were growing up in more analog and static environments. Therefore, this necessitates an updated guideline for ‘family strength’ that differs from the more analog and static environments of previous generations.

The are two research questions, which include the following: (1) What are the factors of family strengths for Generation Alpha in the Thai context? and (2) What are the key dimensions of the measurement model for family strengths for Generation Alpha in the Thai context? There are two objectives of this research, which include the following: (1) to explore the factors of family strengths for Generation Alpha in the Thai context; and (2) to develop a measurement model of family strengths for Generation Alpha in the Thai context.

## 2. Literature Review

### 2.1. Strength-Based Approach

The strength-based approach, rooted in the theory of social work, emphasizes self-determination and individual strengths, and has since been widely adopted across various disciplines [24]. This perspective posits that individuals are inherently resourceful and resilient, even in the face of adversity. By encouraging individuals to recognize and leverage their strengths, this approach fosters a positive self-image, shifting the focus away from negative attributes [15]. Grounded in a humanistic framework, it advocates for human beings to be treated with respect and dignity, acknowledging their capacity to make rational decisions for themselves and effectively navigate crises. This theory is predicated on the belief that that humans are inclined towards goodness and embody essential values in the world [25]. By transitioning the focus from problems to strengths, the strength-based approach recognizes families as essential partners in their success, honoring their knowledge and equipping them to chart their path forward [26].

### 2.2. Family Strengths

The concept of family strengths refers to the qualities and characteristics that enable families to thrive and effectively overcome challenges. It is not about achieving a flawless family dynamic, but rather about cultivating an environment where family members feel loved, supported, and empowered to reach their full potential [27]. In the Thai context, family strength refers to the attributes that contribute to a family’s pursuit of happiness [28]. Key factors of family strengths include commitment, communication, cohesion, coping, and competence.

Commitment refers to the ability of family members to engage in behaviors that promote the well-being of the family unit. Communication involves interacting positively and constructively with one another. Cohesion is the capacity to stick together in both challenging and favorable circumstances. Coping is the ability to deal effectively with adverse life events. Competence reflects the ability to obtain informal and formal support systems and resources to meet family needs or achieve desired goals [29].

The concept of family strengths and the Sustainable Development Goals (SDGs) are not interrelated. Although the SDGs do not explicitly address family strengths, several goals directly pertain to the enhancement of family units, particularly SDGs 1, 3, and 4. Family strengths play a vital role in achieving objectives related to poverty reduction, good health and well-being, as well as quality education for children [30].

### 2.3. Generation Alpha

Generation Alpha encompasses individuals born between 2010 and 2025. Currently, this group consists of children aged 0–13 years, who have been immersed in technology from birth. This environment fosters not only high levels of intelligence but also confidence in expressing their opinions and adeptness in navigating information and technology [5]. Emerging in an era defined by groundbreaking technologies such as AI, the Internet of Things, Generation Alpha is poised to be the most technologically literate generation to date.

While technology presents significant opportunities for learning and connection, its influence on the family dynamics of Generation Alpha warrants careful consideration. Striking a balance between harnessing the positive aspects of technology while mitigating potential downsides should be a primary focus in cultivating healthy family environments for this digital-first generation [31,32].

## 3. Methodology

This research focused on exploring the theoretical underpinnings of the strength-based approach, family strengths, and Generation Alpha. The researchers engaged in a comprehensive analysis and synthesis to develop questionnaire items specifically addressing family strengths in the context of Generation Alpha in Thailand. Factor analysis was utilized in this study, a statistical technique that extracts patterns from large datasets by identifying the groups of variables that share similar traits. Exploratory Factor Analysis (EFA) was utilized to reveal the hidden factors within a dataset without any preconceived assumptions. On the other hand, Confirmatory Factor Analysis (CFA) was conducted to assess the effectiveness of the measurement models by specifying the number of factors and their direct relationships [33].

### 3.1. Populations and Samples

The sample comprised parents and caregivers of Generation Alpha adhering to the sample size criteria proposed by Comrey and Lee [34]. According to their guidelines, a sample size 50 is deemed very poor, 100 is poor, 200 is fair, 300 is good, 500 is very good, and 1000 is considered excellent for the purposes of factor analysis. This research used sample sizes to perform an exploratory factor analysis of about 500 samples and a confirmatory factor analysis of about 500 samples using stratified random sampling from 200, from central, north, south, west, and northeast regions. This research used a total of 1000 samples and collected an extra 10% to prevent inaccurate data responses. Ultimately, the final sample sizes consisted of 521 participants for the exploratory factor analysis and 508 for the confirmatory factor analysis.

### 3.2. Data Collection

Data were gathered using a five-point Likert scale questionnaire. To ensure content validity, the questionnaire was reviewed by three experts in psychology and child development, resulting in an index of Item-Objective Congruence (IOC) of 0.60–1.00. A pilot test was conducted with 50 parents, followed by a discrimination analysis revealing total correlations between 0.280 and 0.896. Two items, V2 and V25, were removed due to correlations below 0.2 score, resulting in a Cronbach’s alpha of 0.952, indicating strong internal consistency.

### 3.3. Data Analysis

The principal analysis method was employed to identify and determine the number of factors, followed by the extraction of common factors using principal axis factoring. An eigenvalue greater than one was considered significant, and the varimax rotation method was applied to enhance the interpretability of the factors. This process allowed for the identification of family strengths specific to Generation Alpha within the Thai context. Subsequently, confirmatory factor analysis was conducted to assess the construct validity of the measurement model of family strengths in this demographic.

### 3.4. Research Ethics

The ethical approval for this research was given by the Institutional Review Board (IRB) of Srinakharinwirot University No. SWUEC-662035, dated 19 October 2023.

## 4. Findings

### 4.1. Results of Exploratory Factor Analysis

The exploratory factor analysis conducted on family strengths for Generation Alpha in the Thai context revealed significant insights. Upon examining the reliability of the data, it was found that the correlation matrix between the question clauses significantly differed from the identity matrix, as indicated by Bartlett’s Test of Sphericity (1473.517, *p* < 0.01). This suggests that the dataset is suitable for use in factor analysis (Kaiser–Meyer–Olkin = 0.914).

The preliminary findings from the survey aimed at quantifying the number of factors associated with family strengths for Generation Alpha revealed five distinct factors, accounting for a cumulative 80.555% of the total variance. Specifically, Factor 1 contributed 21.081%, while Factors 2, 3, 4, and 5 accounted for 17.637%, 14.984%, 14.494%, and 12.358%, respectively. The commonalities for these factors ranged from 0.520 to 0.964, as illustrated in Table 1.

The analysis identified five latent variables representing family strengths for Generation Alpha within the Thai context consisted of five factors, based on factors with an eigenvalue greater than 1.5:

Factor 1, positive relationships, consists of six observable variables, each with a factor-loading value greater than 0.5, as suggested by Laura J. Burton and Stephanie M. Mazerolle (2011) [35]. These variables include the following: Spending quality time with my children (V8), Building positive relationships (V7), Don’t expect too much (V11), Communicating positively (V21), Understanding Gen Alpha children (V6), and Accepting the differences (V22).

Factor 2, enhancing positive attributes, is comprised of five observable variables, including Raising my children to be virtuous (V26), Disciplining my children (V23), Teaching my children to think positively (V4), Encouraging my children to think critically (V10), and Teaching children to be patient (V5).

Factor 3, roles and responsibilities, includes four observable variables, including Understanding parental responsibilities (V9), Bringing up children with care (V16), Being a role model for my children (V14), and Supporting my children (V30).

Factor 4, technology and media literacy, consists of three variables, including Adapting to technology and media (V28), Ensuring my children access to safe and useful media (V27), and Encouraging media literacy (V29).

Factor 5, mental immunity, has two observable variables, including Learning to manage problems (V31) and Mental immunity to facing obstacles (V18).

### 4.2. Results of Confirmatory Factor Analysis

The results of the confirmatory factor analysis of the family strengths for Generation Alpha in the Thai context after the model adjustment were consistent with empirical data (Figure 1), with the value Chi-Square = 442.04 df = 161 and CMIN/df. = 2.75 < 3 and statistics with a Comparative Fit Index (CFI) of 0.93 > 0.90, a Goodness-of-Fit Index (GFI) of 0.92 > 0.90, an Adjusted Goodness-of-Fit (AGFI) of 0.90, a Root Mean Square Error of Approximation (RMSEA) of 0.076 < 0.08, a Root Mean Square Residual (RMRS) of 0.067 < 0.08, a Normed fit index (NFI) of 0.93 > 0.90, and an Incremental Fit Index (IFI) of 0.94 > 0.90.

The first-order confirmatory factor analysis of the family strengths for Generation Alpha in the Thai context consisted of five factors; it found the coefficient between 0.52 and 0.97 (More 0.40) and had a squared coefficient of multilateral correlation (R2) between 37.0 and 95.0%, which considers the quality of the model. It found a convergent validity from the average extracted variance of variables. The Average Variance Extracted (AVE) equaled 0.46–0.81. If there were more models at 0.50 indicates that the measurement model has a good convergence validity. Construct reliability (CR) was equal to 0.83–0.90, more than 0.70, indicating high structural confidence. 

The results of the second-order confirmatory factor analysis of family strengths for Generation Alpha in the Thai context were found to have a coefficient between 0.44 and 0.92 (greater than 0.40) and a squared multi-correlation coefficient (R2) between 19.0 and 85.0%. Convergent validity from Average Variance Extracted (AVE) of 0.78 greater than 0.50 indicates that the measurement model has good convergent accuracy, and construct reliability (CR) of 0.98, greater than 70, indicates high structural confidence. The details are shown in Table 2.

From Table 2, the measurement model of the family strengths for Generation Alpha in the Thai context consists of five factors. All factor loadings are statistically significant at a level of 0.01. The factor with the highest factor loading is a positive relationship factor loading of 0.92, followed by technology and media literacy, parental roles, good attributes, and mental immunity, containing a factor loading value of 0.80, 0.74, 0.55 and 0.44, respectively.

In terms of the measurement model of positive relationships, it was found that it can be measured with six observational variables, with a statistically significant factor loading at a level of 0.05. The indicators of six observational variables can serve as a good indicator of latent variables of positive relationships. The observable variable with the highest factor loading is Building positive relationships (R2) and Understanding Generation Alpha children (R5), with a factor loading of 0.79. Secondly, Accepting the differences (R6), Communicating positively (R4), Spending quality time with my children (R1), and Not expecting too much (R3) contained factor loading values of 0.72, 0.65, 0.54 and 0.52, respectively.

For a measurement model of good attributes, it was found that it can be measured with five observational variables, with factor loading at a statistically significant level of 0.5. The indicators of six observational variables can serve as a good indicator of the latent variables of good attributes. The observable variable with the highest factor loading is Raising my children to be virtuous (A1), which contains a factor loading value of 0.93. Secondly, Teaching my children to think positively (A3), Teaching my children to be patient (A5), Encouraging my children to think critically (A4) and Disciplining my child (A2) contain the factor loading value 0.82, 0.77, 0.72 and 0.71, respectively.

A measurement model of parental roles found that it can be measured by four observational variables with factor loading. All of them are statistically significant at a level of 0.05. The indicators of four observational variables can serve as a good indicator of latent variables of parental roles. The observable variable with the highest factor loading is Supporting my children (R4), which contains a factor loading value of 0.89, followed by Being a role model for my children (Ro3), Understanding parent responsibilities (Ro1) and Bringing up my children with care (Ro2), containing a factor loading value of 0.80, 0.79 and 0.53, respectively.

A measurement model of technology and media literacy found that it can be measured with three observational variables with factor loading at a statistically significant level of 0.5. The indicators of three observational variables can serve as a good indicator of the latent variables of technology and media literacy. The observable variable with the highest factor loading is Encouraging my children to have media literacy (T3), which contains a factor loading value of 0.92. Secondly, Ensuring my children have access to safe and useful media (T2) and Adapting to technology and media (T1) contain a factor loading value of 0.87 and 0.82, respectively.

For a measurement model of mental immunity, it was found that it can be measured with two observational variables with factor loading at a statistically significant level of 0.05. The indicators of three observational variables can serve as a good indicator of the latent variables of mental immunity. The observable variable with the highest factor loading is Mental immunity to facing obstacles (I2), containing a factor loading value of 0.97. This is followed by Learning to manage problems (I1), containing a factor loading value of 0.82.

## 5. Discussion

The measurement model of family strengths for Generation Alpha within the Thai context encompasses five key factors, with the highest factor loading observed in the area of positive relationships, rated at 0.92. This model effectively incorporates six indicators, all demonstrating statistically significant factor loadings at the 0.05 level. The indicators with the highest factor loadings include the following: Building positive relationships (R2), Understanding Gen Alpha children (R5), Accepting the differences (R6), Communicating positively (R4), Spending quality time with my children (R1), and Not expecting too much (R3).

This model aligns with a strength-based approach rooted in humanistic principles, emphasizing the need for human beings to be treated with respect and dignity [36]. Family members possess the autonomy to make rational decisions and can effectively navigate challenges and crises [37]. In Western contexts, family strengths typically encompass factors such as commitment, communication, cohesion, coping, and competence [29]. These qualities contribute significantly to the emotional health and well-being of the Western family [38]. Specifically, in the context of at-risk adolescents in the United States, family strengths consist of family closeness, support, importance, loyalty, protection, love, and responsiveness to health needs [39]

Positive relationships within the family are essential, because Generation X or Millennials often emphasized traditional family values, face-to-face communication, and direct engagement in a less-digitized world. For Generation Alpha, managing screen time, building digital literacy, and keeping strong family connections despite digital distractions are crucial, challenges that earlier generations did not face as much. This finding is consistent with previous research [40], which highlights the enduring and impactful nature of family relationships with respect to well-being throughout the lifespan [4]. Families provide love, support, and a sense of belonging, which are essential for individual and collective well-being. Effective family relationships are characterized by mutual awareness, active listening, and acceptance, which manifest in shared time, assistance, open communication, and joint activities, ultimately fostering a harmonious family environment.

Additionally, the generation gap—the divergence in beliefs, values, and experiences across different age groups—is a phenomenon present in nearly every society. Therefore, it is essential for family members from various generations to embrace their differences. This is consistent with studies by Nagy and Kölcsey [31], which emphasize the importance of acceptance and understanding within families, particularly regarding Generation Alpha. In contemporary Thailand, families increasingly prefer smaller households, often resulting in only one child per family, due to economic considerations. This dynamic leads parents to spend excessive time with their children, coupled with heightened expectations. This is in line with the research by Soratana [41], which highlighted the transition from traditional extended families, where communal child-rearing was common, to smaller units focused on earning a living, often leaving children in the care of others and consequently diminishing familial intimacy.

Another critical factor identified is technology and media literacy, which contains the factor loading value of 0.80. Generation Alpha children are predicted to spend a significant portion of their time using technology, which aligns with the study by Surinya [42] about media and information technology being an important fundamental factor in life and a major contributor to the development of children. Furthermore, parental roles emerged as a significant factor in family strengths for Generation Alpha. This factor is one of the most enduring factors of behavioral science research and has been considered paramount in child development [43]. According to Berrick and Altobelli [44], parents who recognize the inherent strengths of their children are more likely to grant them autonomy and provide the support necessary for reaching their full potential. On the other hand, when children lack the opportunity to participate in family decision-making, they may experience insecurity or a sense of deviance because they are unable to recognize their values or strengths [45]

Good attributes are also a significant factor of family strengths for Generation Alpha. These attributes encompass personal qualities or character traits that are valued in Thai society, such as virtue, patience, and positive thinking. This study is aligned with many studies about the personality of Thai people [46]. Vichit-Vadakan (2011) [47] elaborates on the cultural emphasis on social harmony in Thailand, where avoidance of confrontation and the use of humor to diffuse tension are common, contrasting sharply with Western values that prioritize individual rights, freedom, and personal responsibility [48] (Pae, 2020).

Finally, mental immunity is also found to be a factor of family strength for Generation Alpha. Thailand is currently grappling with significant psychological issues, including depression and anxiety, making the promotion of self-care and emotional regulation essential in contemporary Thai society, particularly among families of Generation Alpha. There are three key processes for promoting family strengths to cope with trauma in America, including family belief systems, organization patterns, and communication processes [49].

## 6. Conclusions

In sum, families serve as the cornerstone of every society. Understanding family strengths for Generation Alpha necessitates recognition of the unique characteristics of this generation, alongside the evolving dynamics of family life. Supporting the positive aspects of family life is crucial for building resilient communities and societies. Thailand places a strong emphasis on cultivating healthy family structures, aiming to enhance social stability and foster positive societal impact. The understanding and leveraging of these potential family strengths can contribute to the well-being and success of Generation Alpha. Families that embrace adaptability, inclusivity, and a positive approach to the unique challenges and opportunities of the digital age can thrive and provide a supportive environment for their children. The benefits of family strengths for Generation Alpha extend far beyond individual households, influencing the well-being of entire communities and contributing to social harmony. These findings expand knowledge about strengthening Generation Alpha families, which is suitable for the future context of Thai society. The practical implications suggest that practitioners, policymakers, researchers, and other stakeholders can utilize this measurement model to devise policies and plans that address family issues, mitigate social problems, and guide the enhancement of future families in Generation Alpha.

## Figures and Tables

**Figure 1 behavsci-14-00921-f001:**
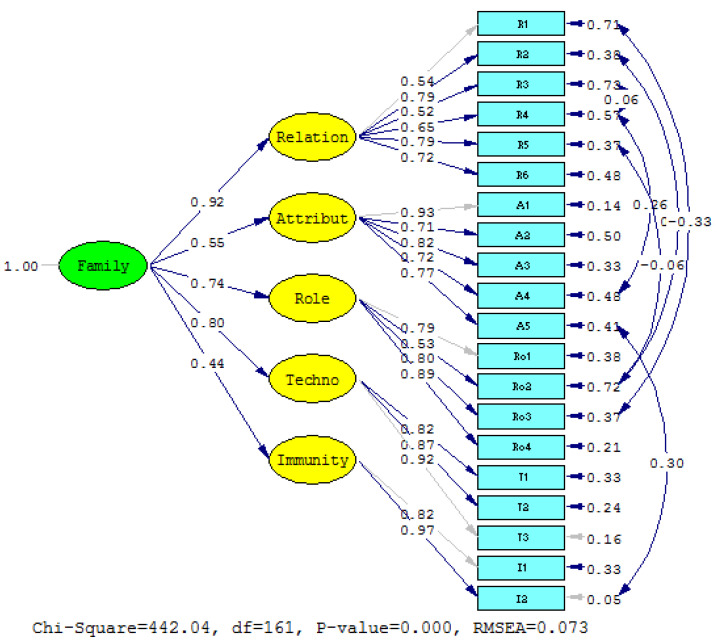
The Measurement Model of Family Strengths for Generation Alpha in the Thai Context.

**Table 1 behavsci-14-00921-t001:** Analysis of Factor Loading of Family Strengths for Generation Alpha.

Items	Factor Loading					Communalities
	Factor 1	Factor 2	Factor 3	Factor 4	Factor 5	
V8	**0.806**	−0.037	0.187	0.264	0.023	0.756
V7	**0.800**	0.037	0.211	0.144	−0.042	0.709
V11	**0.765**	0.233	0.200	0.126	0.145	0.716
V21	**0.646**	0.397	0.362	0.330	−0.230	0.869
V6	**0.641**	0.094	0.299	0.073	−0.071	0.520
V22	**0.534**	0.206	0.498	0.410	−0.078	0.868
V3	0.417	0.405	0.507	0.323	0.049	0.909
V20	0.408	0.082	−0.043	0.417	0.392	0.706
V13	0.403	0.386	−0.019	0.262	0.434	0.769
V19	0.341	0.020	0.466	−0.098	0.523	0.792
V26	−0.043	**0.932**	0.149	0.258	−0.068	0.964
V23	−0.128	**0.817**	0.379	−0.250	0.114	0.903
V4	0.145	**0.784**	0.214	0.166	−0.143	0.729
V10	0.535	**0.728**	0.289	0.107	0.051	0.913
V5	0.351	**0.587**	−0.092	0.181	0.335	0.749
V12	0.226	0.452	0.063	0.386	−0.244	0.688
V9	0.257	0.046	**0.835**	0.170	0.120	0.808
V16	0.318	0.260	**0.712**	0.072	0.137	0.699
V14	0.369	0.211	**0.644**	−0.049	0.098	0.608
V30	−0.080	0.229	**0.542**	0.051	0.639	0.883
V1	0.205	0.076	0.421	0.416	0.417	0.781
V28	0.351	0.141	0.173	**0.774**	0.283	0.852
V27	0.293	0.568	0.012	**0.731**	−0.060	0.946
V29	0.345	0.442	0.134	**0.697**	0.302	0.908
V17	0.463	0.203	0.322	0.486	0.210	0.873
V24	0.458	0.250	0.080	0.414	0.530	0.937
V15	0.313	0.444	0.513	0.330	0.221	0.889
V32	0.276	0.288	0.492	0.296	−0.360	0.777
V31	−0.133	−0.051	0.190	0.125	**0.911**	0.888
V18	0.127	−0.175	0.092	0.014	**0.810**	0.757
Eigenvalues	13.577	3.786	2.734	2.218	1.851	
% of Variance	21.081	17.637	14.984	14.494	12.358	

**Table 2 behavsci-14-00921-t002:** Results of Confirmatory Factor Analysis of the Family Strengths for Generation Alpha in the Thai Context.

Factors (Latent Variables)	Observable Variables	β	B (SE)	t	R^2^	AVE	CR
First-order confirmatory factor analysis							
Positive Relationship	Spending quality time with my children (R1)	0.54	0.35	-	0.39	0.46	0.83
	Building positive relationships (R2)	0.79	0.57 (0.04)	12.89 *	0.62		
	Don’t expect too much (R3)	0.52	0.25 (0.03)	9.61 *	0.37		
	Communicating positively (R4)	0.65	0.33 (0.03)	11.84 *	0.43		
	Understanding Gen Alpha (R5)	0.79	0.73 (0.06)	12.97 *	0.63		
	Accepting the differences (R6)	0.72	0.42 (0.03)	12.23 *	0.84		
Good Attributes	Raising my children to be virtuous (A1)	0.93	0.63	-	0.86	0.63	0.89
	Disciplining my children (A2)	0.71	0.45 (0.02)	19.94 *	0.50		
	Teaching my children to think positively (A3)	0.82	0.68 (0.03)	25.94 *	0.67		
	Encouraging my children to think critically (A4)	0.72	0.53 (0.02)	21.95 *	0.52		
	Teaching my children to be patient (A5)	0.77	0.44 (0.01)	30.18 *	0.59		
Parental Roles	Understanding parental responsibilities (Ro1)	0.79	0.77	-	0.62	0.52	0.86
	Bringing up my children with care (Ro2)	0.53	0.39 (0.03)	12.36	0.38		
	Being a role model for my children (Ro3)	0.80	0.53 (0.03)	19.70	0.63		
	Supporting my children (Ro4)	0.89	0.43 (0.02)	21.65	0.79		
Technology and Media Literacy	Adapting to technology and media (T1)	0.82	0.82 (0.03)	25.62 *	0.67	0.76	0.90
	Ensuring my children have access to safe and useful media (T2)	0.87	0.95 (0.04)	29.49 *	0.76		
	Encouraging my children to have media literacy (T3)	0.92	1.00	-	0.84		
Mental Immunity	Learning to manage problems (I1)	0.82	0.44	-	0.67	0.81	0.89
	Having mental immunity to face obstacles (I2)	0.97	0.96 (0.03)	30.91 *	0.95		
Second-order confirmatory factor analysis							
Strengthening the Alpha Generation Family	Positive relationships	0.92	0.92 (0.07)	12.54 *	0.85	0.78	0.98
	Good attributes	0.55	0.55 (0.05)	11.86 *	0.30		
	Parental Roles	0.74	0.74 (0.05)	14.48 *	0.55		
	Technology and media literacy	0.80	0.80 (0.02)	18.22 *	0.65		
	Mental immunity	0.44	0.44 (0.05)	9.64 *	0.19		

Note: * *p* < 0.01. SE and T-values are not reported as constrained parameters.

## Data Availability

Data are unavailable due to privacy or ethical restrictions.

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
