# Peer review of "The Measurement Model of Family Strengths for Generation Alpha in the Thai Context"

_behavsci, 2024, doi:10.3390/bs14100921_

Round 1

Reviewer 1 Report

Comments and Suggestions for Authors

Thank you for inviting me to review this paper.

My comments are as below:

1.      Abstract - The abstract should consist of several elements, namely the title statement, research problem, research objective, methodology and research findings. The author, however, did not highlight the problem statement.

            2. Page 2, Line 48 - It is better to choose one citation writing method  either using Endnote or APA style

3.      I suggest that the discussion be separated from the conclusion. This is because the discussion is carried out in depth while the conclusion can be done in one paragraph only.

4.      The Interpretation of the results is fascinating.

Thank you.

Comments on the Quality of English Language

No comment

Author Response

Dear Reviewer 1

Thank you very much for your valuable review of our paper. We truly appreciate your kind support. We have revised the paper according to your comments as follows:

  1. Abstract - The abstract should consist of several elements, namely the title statement, research problem, research objective, methodology and research findings. The author, however, did not highlight the problem statement.

We have highlighted the problem statement as follows:

The concept of Thai family has shifted from the traditional family to families of various forms, such as single-parent families, same-sex families, skipped-generation families, and one-person households which affects not only mental health but future relationships of the children.

  1. Page 2, Line 48 - It is better to choose one citation writing method  either using Endnote or APA style

We have changed it to Endnote according to the journal format.

  1. I suggest that the discussion be separated from the conclusion. This is because the discussion is carried out in depth while the conclusion can be done in one paragraph only.

We separated the discussion from a conclusion.

  1. The Interpretation of the results is fascinating.

Thank you very much again. 

Reviewer 2 Report

Comments and Suggestions for Authors

This paper deals with an interesting and important topic, but I see an urgent need for improvement in some points:

1) The authors owe the reader a more thorough explanation as to why their approach applies specifically to ‘Generation Alpha’ and not to other generations before it. To what extent do the guidelines for the ‘family strength’ of Generation Alpha differ from the guidelines for other (earlier) generations? (see l. 76f, 289f.).

2) The chapter "findings" is written far too technically, focussing primarily on methodological aspects (factor analysis etc.) instead of clearly stating the hypotheses guiding the research. Two simple statements about the research objectives without formulating precise research hypotheses in which the selection of the variables used is justified are not enough. (l. 79-81). What can and cannot be deduced from the so-called findings? The authors' primary concern in this chapter should not be to show that they have understood the methods used, but rather the substantive aspects of their research. 

3) Clarifications are also necessary in the Methodology chapter. Where and when was the data collected, by whom, who was surveyed, only parents and guardians or also members of Generation Alpha? Particularly in the area of ‘family’, there are often considerable differences between rural and urban areas, possibly also gender-specific differences. If these differences were not addressed in the study, they should at least be pointed out.

4) A short section on the limitations of the study is missing. A short section on the limitations of the study is missing, as is a more critical analysis of the family strength approach used. At the very least, the problems of the idealisation of the nuclear family, the influence of economic and structural determinants and the tension between ‘internal resources - external solutions’ should not go unmentioned.

Comments on the Quality of English Language

Moderate language editing is required, especially with regard to incomplete sentences.

Author Response

Dear Reviewer 2,

Thank you very much for your valuable time reviewing our manuscript. We have revised our manuscript according to your kind suggestion as follow:

Reviewer 2

  1. This paper deals with an interesting and important topic, but I see an urgent need for The authors owe the reader a more thorough explanation as to why their approach

applies specifically to ‘Generation Alpha’ and not to other generations before it. To what extent do the guidelines for the ‘family strength’ of Generation Alpha differ from the guidelines for other (earlier) generations? (see l. 76f, 289f.).

improvement in some points:

The research can be used as a guideline for family strength for Generation Alpha in the Thai context who growing up in a world of advanced artificial intelligence, digital connectivity, and dynamic environment which significantly influences their socialization and family dynamics differently from previous generation which growing up in more analog and static environments. Therefore, it is necessitate an updated the guidline for ‘family strength’ that differs from the more analog and static environments of previous generations (McCredle and Fell, 2020)

Positive relationships within the family are essential because Generation X or Millennials, often emphasized traditional family values, face-to-face communication, and direct engagement in a less digitized world. For Generation Alpha, managing screen time, building digital literacy, and keeping strong family connections despite digital distractions are crucial, challenges that earlier generations didn’t face as much (Garbe et al., 2020)

2) The chapter "findings" is written far too technically, focussing primarily on methodological aspects (factor analysis etc.) instead of clearly stating the hypotheses guiding the research. Two simple statements about the research objectives without formulating precise research hypotheses in which the selection of the variables used is justified are not enough. (l. 79-81). What can and cannot be deduced from the so-called findings? The authors' primary concern in this chapter should not be to show that they have understood the methods used, but rather the substantive aspects of their research. 

Results of Exploratory Factor Analysis  It is used to answer the 1st objective-to explore the factors of family strengths for Generation Alpha in the Thai context to find out the number of strong elements of the Alpha Generation family in the Thai context Measurement model Using statistics. Exploratory Factor Analysis

Results of Confirmatory Factor Analysis   It is used to answer the purpose of the 2 to develop a measurement model of family strengths for Generation Alpha in the Thai context. To confirm that: Measurement model Obtained from the objective 1 is appropriate (Fit)

3) Clarifications are also necessary in the Methodology chapter. Where and when was the data collected, by whom, who was surveyed, only parents and guardians or also members of Generation Alpha? Particularly in the area of ‘family’, there are often considerable differences between rural and urban areas, possibly also gender-specific differences. If these differences were not addressed in the study, they should at least be pointed out.

This research  collects demographic data as follows:

  1. Status (Parents, Guardians)
  2. You have a child/child in your care who is a Generation Alpha child. ...... people
  3. You have a child/child in your care who is a Generation Alpha child. What age range and what level of study?

   Children 0-3 years old (preschool)

   Children 4-6 years old (Early Childhood)

   Children 7-13 years old (Primary school children)

However, because what needs to be studied is a model for measuring the strength of the Alpha Generation family in the Thai context. This does not require demographic data  of the sample to analyze the measurement model.

 In addition, the questionnaire asked parents and guardian whether V1-V32 (obtained from the review) is an indicator of the strength of the Alpha Generation family in the context of Thai society. Therefore, the researcher did not consider that demographics have an effect or do not want to compare between demographics, so they did not collect data on gender, age, education, domicile, etc. However, this suggestion will be used by the researcher for further research.

4) A short section on the limitations of the study is missing. A short section on the limitations of the study is missing, as is a more critical analysis of the family strength approach used. At the very least, the problems of the idealisation of the nuclear family, the influence of economic and structural determinants and the tension between ‘internal resources - external solutions’ should not go unmentioned.

The limitations of this research is factor analysis assumes that the relationships between variables are linear. If non-linear relationships exist, the results might not accurately represent the data structure.

In addition, we have asked native writers to revise our grammar and punctuation.

Reviewer 3 Report

Comments and Suggestions for Authors

The research purpose is clearly and and an appropriate method of study is applied. Also prospects for futher research are provieded. However there are some minor points, whice author should consider in order to improve the paper. 

-Specify the gap in the literature review.

-The research questions should be clearly stated.

-Better cohensive connection between discussion and research findings should be made.

-it is necesseray to enrich with more extra  contemporary references. 

Author Response

Dear Reviewer 3,

Thank you very much for your valuable time reviewing our manuscript. We have revised our manuscript according to your kind suggestion as follows:

The research purpose is clearly and and an appropriate method of study is applied. Also prospects for futher research are provieded. However there are some minor points, whice author should consider in order to improve the paper. 

-Specify the gap in the literature review.

We have highlighed gap in the literature reviews on line 70-77  “According to the literature review, the factor of family strengths for Generation Alpha has not been studied in the Thai context. There were only a few studies about family strength, namely the factors of family strength [19], indicators of family strength [20], indicators of strong families [21], second order components of Thai family strength indi-cators [22] and the construction of family strength factors [23]. Therefore, this research aims to explore factors and develop a measurement model of family strengths for Generation Alpha in the Thai context. The results of this research can be used as a guideline for family strength for Generation Alpha in the Thai context.”

-The research questions should be clearly stated.

The are two research questions, which include the following: (1) What are the factors of family strengths for Generation Alpha in the Thai context; and (2) What are the key dimensions of the measurement model for family strengths for Generation Alpha in the Thai context.

-Better cohensive connection between discussion and research findings should be made.

We revised the manuscript to better connection between discussion and research findings.

-It is necesseray to enrich with more extra  contemporary references.

We have added more contemporary references.

 In addition, we have asked a native writer to check grammar and punctuation for us.

Thank you so much once again

Round 2

Reviewer 2 Report

Comments and Suggestions for Authors

thank you for the revision, no more objections